# Early Life Nutrition and Its Effects on the Developing Heifer: Immune and Metabolic Responses to Immune Challenges

**DOI:** 10.3390/ani15101379

**Published:** 2025-05-10

**Authors:** Emma M. Ockenden, Victoria M. Russo, Brian J. Leury, Khageswor Giri, William J. Wales

**Affiliations:** 1Agriculture Victoria, Ellinbank, VIC 3821, Australia; 2The Faculty of Science, The University of Melbourne, Parkville, VIC 3010, Australia; 3Centre for Agricultural Innovation, The University of Melbourne, Parkville, VIC 3010, Australia; 4Agriculture Victoria, Bundoora, VIC 3083, Australia

**Keywords:** immune challenge, increased milk feeding, dairy calf immunity, dairy heifer resilience

## Abstract

Combinations of pre- and postweaning nutritional strategies were used to determine the effect early life nutrition has on the future resilience of dairy replacement heifers. Positive influences of increased milk feeding on calf immune responses and metabolic characteristics were observed in the preweaning phase. However, while somewhat apparent, the longer-term advantages of increased preweaning nutrition on immunity need to be confirmed. These results may lead to new early-life management strategies that reduce the high incidence of young replacement stock losses, improving the welfare, economic and environmental impacts of the Australian dairy industry.

## 1. Introduction

Rearing replacement heifers is one of the largest investments on a dairy farm and focuses on producing resilient replacement cows with a high milk production potential [1]. In Australia, calf morbidity and mortality rates are approximately twice the industry targets (morbidity 23.8% with <10% target and mortality 5.6% with <3% target) [2], with approximately 30% of replacement stock born in Victoria, Australia exiting the herd before their second calving [3]. Veterinary related costs further add to the overall high costs associated with rearing replacement stock [2]. More resilient cows suffer fewer metabolic or pathogenic diseases, and in turn, their production performance is less likely to be impacted. These resilience characteristics rely on physiological systems including immunity and metabolism [4]. Improving calf and heifer rearing strategies has the potential to increase their resilience. Increasing the resilience of heifers not only improves the health, welfare and productivity of the herd, but also reduces animal wastage, increases profitability and maintains the social licence to operate the industry [4,5].

Early life nutrition is widely accepted to influence the physiological development of mammals [6]. However, the effect preweaning nutrition has on the short- and long-term immune and metabolic development of calves is contradictory within the literature. Furthermore, the effect of postweaning growth rates on the development of these systems is relatively unknown. The innate immune system is one of the body’s first lines of defence (after physical and mucosal barriers) and is very general, recognising cells as either self or non-self [7]. This system includes but is not limited to the various types of white blood cells (neutrophils, monocytes, lymphocytes, basophils and eosinophils) that identify and destroy invading pathogens [8]. The literature regarding the influence of increased preweaning nutrition on these cell types is unfounded, with experiments finding it beneficial [9,10], without difference [11], and in some instances detrimental [12,13]. There are links associating early life nutrition and the gut microbiome to long-term immune status [14,15]. Recent work has also reported preservation of superior immune status at 12 months of age from increased milk feeding as a calf [16]. However, the effect postweaning growth rate has on the immune system development of heifers, or if this previously observed preservation from the preweaning phase can be further manipulated by postweaning growth rate, is unknown.

Preweaning nutrition is also reported to influence the metabolic development of calves. Increasing concentrations of beta-hydroxybutyrate (BHB) and non-esterified fatty acids (NEFA) are associated with negative energy balance in adult cows [17,18]. However, the effect of increased BHB in calves is unclear [19,20]. While feeding increased volumes of milk provides greater amounts of energy via glucose [21,22], with subsequent increased concentrations of insulin-like growth factor (IGF-1) with the associated increased growth rates [23,24], there are also perceived disadvantages regarding the insulin sensitivity of these animals, indicated by insulin sensitivity index calculations such as QUICKI [25,26,27]. The long-term impact of pre- and postweaning nutrition on the baseline metabolic characteristics of these heifers has been previously reported in a companion paper [28]. Results from this experiment indicate that the metabolic characteristics are influenced by the current level of nutrition, independent of previously applied preweaning treatment [28]. These baseline results have been used in the current paper in conjunction with new response data to investigate the effect the nutritional strategies are having on the metabolic response capabilities of these heifers when challenged. Heifers that are more resilient are less likely to be impacted by disturbances to homeostasis [4]; therefore, superior responses or less negative effects on the metabolic biomarkers could be expected from those heifers on a higher level of nutrition and verified by investigation.

This experiment aimed to identify differences in immune and metabolic biomarkers in heifers reared at four different planes of nutrition, including a combination of either a high or low preweaning treatment (8 L vs. 4 L milk daily) with a high or low postweaning growth rate (pre-calving target bodyweight of 600 kg vs. 530 kg). Heifer resilience was assessed using immune challenges. The hypotheses tested were (1) that calves that received a high milk allowance in the preweaning phase will have physiologically superior immune and metabolic biomarkers following an immune challenge than calves that received a low milk allowance in the preweaning phase, (2) that this superior immune response will still be evident at 8 and 13 months of age regardless of postweaning growth rates and (3) that the metabolic response at 8 and 13 months of age will be dependent on the current postweaning growth rate and not effected by the previously applied preweaning nutritional treatment.

## 2. Materials and Methods

This experiment was made up of two parts. Part A focused on the production effects and baseline metabolism of different pre- and postweaning growth rates of dairy heifers [28], while Part B (current paper) investigated the immune and metabolic responses to several immune challenges imposed on the same dairy heifers at various stages of their development (6 weeks, 8 months and 13 months of age). Animal nutritive intakes and growth data are fully outlined in Ockenden et al. [28].

The preweaning phase of this experiment was carried out at the Agriculture Victoria Dairy Research Centre, Ellinbank, Victoria, Australia (38°14′ S,145°56′ E) from July to October 2021. The postweaning phase was undertaken at Agriculture Victoria’s heifer rearing contractor property in Trafalgar, Victoria, Australia. Experimental protocols were approved by the Agricultural Research and Extension Animal Ethics Committee, application number 2021-08. All experimental protocols were conducted in accordance with the Australian Code of Practice for the Care and Use of Animals for Scientific Purposes [29].

### 2.1. Experimental Design and Treatments

Eighty Holstein Friesian replacement heifers were enrolled at birth (Australian Spring) until approximately 20 months of age. One of four treatments was randomly assigned to calves at birth: high–high (HH), high–low (HL), low–high (LH) and low–low (LL) (treatment titles signify the preweaning treatment to postweaning growth rate). Treatments were balanced for birthweight, parity of the dam and estimated Balance Performance Index (BPI) (Australian national selection index including economic and genetic traits such as production, health, fertility, longevity and feed efficiency). Estimations were calculated prior to birth using the average of the parents’ BPI. Balanced parameters were checked as treatments neared full allocation, and in instances where parameters were skewed between treatments, randomisation was overruled.

During the preweaning phase, 40 heifers received either of the following:High preweaning treatment—8 L of milk daily (HH & HL)Low preweaning treatment—4 L of milk daily (LH & LL)

At birth, calves were separated into four separate pens sequentially by date of birth to allow for age-appropriate grouping (20 calves per pen). Each pen contained five calves from each of the four treatment groups, and each pen followed its own timeline. All treatment allocations were full within 22 days.

At weaning, heifers then followed either their predetermined high (HH and LH) or low (HL and LL) postweaning growth rate towards one of two final pre-calving weights (600 kg vs. 530 kg), giving rise to the four experimental treatment groups of 20 heifers each (HH, HL, LH and LL). Postweaning treatments were defined by set target weights based on current Australian heifer rearing strategies, with an ideal target weight of 85% mature liveweight at first calving. The range in treatment target bodyweights was determined using herd historical data from the Ellinbank Research Dairy Farm. Essential bodyweight data have been summarised from Ockenden et al. [28] in Table 1.

### 2.2. Animal Management

Heifers were removed from their dams within 8 h of birth and fed 4 L pooled fresh colostrum with >22% Brix index. Successful passive transfer of immunity was confirmed using serum Brix refractometer methods prior to enrolment in the experiment [30]. Only calves with a serum Brix index concentration at or above 8.1% (equivalent to ≥10 g/L serum IgG) were included in the experiment [31]. The first four milk meals post colostrum feeding were 2 L of transitional milk (milk from the 2nd to the 4th milking post birth) via individual teat feeders within nursery pens before the calves were transferred to the experimental calf-rearing facility. In the experimental calf-rearing facility, heifers were group-housed within four pens (twenty heifers per pen), and each pen contained five heifers assigned to each of the four treatment groups. Heifers were fed saleable whole milk with individual intakes recorded via automatic calf feeders (Lely Calm; Lely Australia, Truganina, Victoria, Australia) as per their treatment allocation. Heifers in the high treatment group were progressively accustomed to the higher milk volume with allocations limited to 6 L for the first day, then 8 L for the remaining experimental period. To avoid overconsumption, each milk meal size was capped at 2 L per feed, with a 2 h break between meals for both treatments. Heifers also had access to up to 3 kg/calf per day of a commercial calf concentrate (Reid Stockfeeds—Gippsland, Trafalgar, Victoria, Australia), which was also fed out, and individual intakes recorded, via automatic concentrate feeders (Lely Calm; Lely Australia, Truganina, Victoria, Australia). Water and lucerne hay (*Medicago sativa*) were available ad libitum within each pen; however, these intakes were not recorded. Bodyweights were initially recorded at 24–48 h post birth (as birth bodyweight), then weekly until weaning using electronic scales. Weaning commenced at 10 weeks of age, and milk allocation was reduced gradually over 10 days. Once weaned, heifers entered the postweaning phase of the experiment.

In the postweaning phase, heifers were run as a single herd, with pasture providing the majority of the diet. Pasture was provided at what is considered ad-libitum (approximately 20 kg DM/cow per day). Individual pasture intakes were not recorded as part of this experiment. A commercial concentrate (Reid Stockfeeds-Gippsland, Trafalgar, Victoria, Australia, Calf weaner until 12 months of age, then Yearling mix from 12 months+) was also available via an in-paddock automatic feeder (Velos Cow Feeding Station, Nedap N.V. Livestock Management, Groenlo, The Netherlands) that recorded daily individual intakes. Weighing occurred fortnightly for the first 6 months at pasture, then monthly thereafter. Bodyweight targets for each treatment group were achieved by either altering the amount of concentrate on offer (i.e., increasing concentrate allocation for treatment groups in instances where additional bodyweight gains were required to achieve target weights) or by restricting pasture by limiting the paddock area after each weighing. In instances where any one treatment was exceeding target, all heifers were restricted pasture using strip-grazing methods. Heifers completed the experiment at 20 months of age.

### 2.3. Immune Challenge

Heifers were subject to three immune challenges throughout their early life using modified protocols of Aleri et al. [32] and Ockenden et al. [9]. The first immune challenge was in the preweaning phase at 6 weeks (±3 days) and was repeated at approximately 8 and then 13 months of age. Each immune challenge was initiated by administering a commercially available registered vaccination for protection against common leptospirosis and clostridial diseases (UltraVac 7in1, Zoetis, Rhodes, Australia) according to the manufacturer’s protocol. Subsequent immune and metabolic responses were compared between treatments via blood samples. In the preweaning phase, access to milk and concentrate was withheld for two hours prior to blood sampling; however, hay was still available within each pen. No fasting was implemented for either postweaning immune challenge. For each challenge, an initial blood sample was taken immediately prior to giving the vaccination, and the second blood sample was taken 10 days later (as per refined protocols established in Aleri et al. [33]). Each blood sample involved withdrawal of 35 mL of blood via jugular venepuncture into four vacutainers (1 × 10 mL lithium heparin, 1 × 10 mL plain, 1 × 10 mL EDTA and 1 × 5 mL fluoride/oxalate tubes). Immediately after sampling EDTA, lithium heparin and fluoride/oxalate vacutainers were stored on ice before being centrifuged at 1578× *g* for 15 min at 4 °C to obtain the supernatant. Prior to centrifugation, a sample of whole blood was extracted from the EDTA tube and sent for white blood cell count analyses within 8 h of sampling. Samples collected in the plain vacutainers were incubated at 24 °C for 2 h prior to centrifugation at 1258× *g* for 15 min at 24 °C to obtain the serum. Plasma and serum samples were then stored at −20 °C prior to analyses.

### 2.4. Biomarker Analyses

Immune biomarkers analysed included a white blood cell count with differential profiling (total white blood cell count (WBC), neutrophils, monocytes, lymphocytes, basophils and eosinophils). Immune biomarkers were analysed from the subsample of whole blood extracted from the EDTA tube mentioned above. These samples were analysed using a Roche Sysmex xn 1000 (Sysmex, Norderstedt, Germany) at a commercial laboratory (U-Vet, Werribee, Victoria, Australia).

Metabolic biomarkers analysed were BHB, NEFA, glucose, insulin and insulin-like growth factor (IGF-1). The BHB, NEFA and glucose assays were conducted on a ChemWell 2910 automated analyser at AgriBio laboratories (Bundoora, Victoria, Australia) using Catachem Inc. reagents, controls and calibrators as per manufacturer’s instructions. Insulin and IGF-1 assays were developed and performed at the Assay Centre, School of Agriculture, Food and Ecosystems Sciences, the University of Melbourne. Insulin concentrations were determined in a homologous, double-antibody radioimmunoassay (RIA). The RIA was conducted using purified insulin antiserum raised in guinea pig (Antibodies Australian) and purified bovine insulin for iodination and standard (Sigma-Aldrich, Darmstadt, Germany, cat#I5500). Complete assay methods can be found in Ockenden et al. [9]. Bovine plasma IGF-1 concentration was measured following sample extraction in a homologous, double-antibody RIA. The RIA was performed using human IGF-1 antiserum raised in rabbit (National Hormone and Peptide Program, (NHPP) AFP4892898) and human IGF-1 for iodination and standard (GroPep, Human IGF-1 (Receptor Grade)). Full assay methods for IGF-1 can also be found in Ockenden et al. [28].

Insulin sensitivity was determined using the quantitative insulin-sensitivity check index (QUICKI) with the following equation [34].QUICKI = 1/[log glucose (mg/dL) + log insulin (mIU/L)]

### 2.5. Statistical Analysis

The effects of treatments on metabolic biomarkers were analysed using general analysis of variance in GenStat 22nd Edition (VSN International, Hemel Hempstead, UK). The treatment structure included main effect of preweaning treatment, main effect of postweaning treatment, main effect of sampling occasion, all two-way interactions and the three-way interaction amongst preweaning treatment, postweaning treatment and sampling occasion. The pen/calf was the blocking structure. The effects of birth bodyweight, estimated BPI and age of the dam were fitted as covariates. The general analysis of variance model can be written in the following equation form:$$y=\mu +\beta_{1}w+\beta_{2}s+\beta_{3}a+m+c+t+mc+mt+ct+mct+P+\varepsilon_{PA}$$
where y is the response variable of interest, *μ* is overall constant (grand mean), w is the birth bodyweight (pre-experiment) with coefficient $\beta_{1}$, s is the estimated BPI score (pre-experiment) with coefficient $\beta_{2}$, and a is the age of dam (pre-experiment) with coefficient $\beta_{3}$. The *m* is the main effect of preweaning treatment (milk), *c* is the main effect of postweaning treatment (concentrate), *t* is the main effect of sampling occasion(time), *mc* is the two-way interaction between preweaning and postweaning diet treatments, *mt* is the two-way interaction between preweaning diet treatment and sampling occasion, *ct* is the two-way interaction between postweaning diet treatments and sampling occasions, *mct* is the three-way interaction amongst preweaning treatment, postweaning treatment and sampling occasion, and *p* is the main effect of pen (blocking factor). The $\varepsilon_{PA}$ is the residual error term, representing the effect of calf (animal) within a pen. The preweaning data were analysed using a simplified version of the above-described model, excluding postweaning treatment effects and their associated interactions. The residuals from the general analysis of variance were assumed to be normally distributed, with zero mean and constant variance. Normality of residuals was checked using histograms and plots of residuals versus fitted values. Insulin (6 weeks), eosinophil (8 months and 13 months) and NEFA (13 months) data were required to be logarithmically transformed to meet the assumption of normality. These treatment means were then back-transformed using the basis correction factor $exp\;(\hat{\mu }+\frac{{\hat{\sigma }}^{2}}{2})$, where $\hat{\mu }$ and ${\hat{\sigma }}^{2}$ are the estimated treatment mean and residual variance, respectively, in the logarithmic scale [35].

One calf from the HH treatment group died shortly after entering the postweaning phase of the experiment; therefore, only her preweaning data are included in the results.

## 3. Results

### 3.1. 6-Week Immune Challenge

At 6 weeks of age, all biomarkers had a significant response to the vaccination (*p* < 0.05) (Figure 1). The vaccination resulted in significant increases (*p* < 0.05) to WBC and lymphocytes, while basophils and neutrophils significantly decreased with no differences between treatments. Monocytes increased with vaccination and were significantly higher for the high preweaning treatment group than the low preweaning treatment group (*p* = 0.017). Eosinophils were also significantly different between preweaning treatments (*p* = 0.008), and counts reduced with vaccination for both treatments. BHB and NEFA concentrations also both reduced with vaccination. BHB concentrations were significantly lower for the high preweaning treatment groups (*p* < 0.001), while NEFA concentrations were higher (*p* = 0.042). Glucose and IGF-1 increased with vaccination and were significantly higher for the high preweaning treatment group than for the low preweaning treatment group (glucose: *p* < 0.001; IGF-1: *p* < 0.001). There was a significant two-way interaction effect between preweaning treatment and vaccination response for insulin (*p* = 0.009), such that insulin concentration decreased with vaccination for high preweaning treatment groups, whereas insulin concentrations for the low treatment groups increased with vaccination. The same significant two-way interaction effect occurred with the QUICKI results (*p* = 0.008), where the high treatment group increased with vaccination, and the low treatment group reduced.

### 3.2. 8-Month Immune Challenge

At 8 months of age, there was no significant effect of treatment or vaccination on WBC, neutrophils, lymphocytes, basophils or NEFA. As shown in Figure 2, monocytes reduced post vaccination for all treatments (*p* < 0.001). Eosinophils also significantly reduced post vaccination for all treatments (*p* < 0.001). Additionally, there was a significant difference in the number of eosinophils between preweaning treatments: the high-milk-fed calves (HH and HL) had a significantly greater eosinophil count than the low-milk-fed calves (LH and LL) at 8 months of age (*p* < 0.001). There was an effect of vaccination on BHB, glucose and IGF-1, such that they all reduced post vaccination for all treatments (BHB: *p* = 0.026; glucose: *p* < 0.001 and IGF-1: *p* < 0.001) There was a two-way interaction effect between postweaning growth rate and vaccination for insulin (*p* = 0.017), such that insulin concentrations did not change from pre- to post-vaccination for the high postweaning groups (HH and LH), yet concentrations reduced significantly post vaccination for the low postweaning treatment groups (HL and LL). There was also a two-way interaction effect for response to the vaccination and postweaning treatment for the QUICKI results (*p* = 0.023). In this instance, treatments (HH, HL and LL) increased with vaccination, but the LH group decreased.

### 3.3. 13-Month Immune Challenge

At 13 months of age, all biomarkers were affected by treatment and/or vaccination (Figure 3). Monocyte, lymphocyte and basophil numbers significantly reduced with vaccination (monocytes: *p* < 0.001; lymphocytes: *p* < 0.001; basophils, *p* = 0.002), with no treatment differences detected. A three-way interaction effect (preweaning treatment x postweaning treatment x vaccination) was evident for WBC (*p* = 0.002), neutrophils (*p* = 0.004) and eosinophils (*p* = 0.010), with similar, seemingly random, differences between treatments observed for all three biomarkers. BHB concentrations significantly increased post vaccination (*p* = < 0.001), and there were also no differences detected between treatments. NEFA concentrations significantly reduced for all treatment groups post vaccination (*p* = 0.020) and were significantly higher for the low postweaning groups (HL and LL) than the high postweaning groups (HH and LH) (*p* = 0.013). Insulin concentrations significantly increased with vaccination for all treatment groups (*p* < 0.001) and were significantly higher in the high postweaning groups (HH and LH) than in the low postweaning groups (LL and HL) (*p* < 0.001). There was an interaction effect of glucose concentrations with postweaning treatment and time, such that the high postweaning groups’ (HH and LH) glucose concentration did not change with vaccination; however, in the low postweaning groups (LL and HL), glucose concentration increased with vaccination (*p* = 0.006). A similar interaction effect was found for the QUICKI results, such that the low postweaning treatments started significantly higher and reduced with vaccination, while there was no change for the high postweaning treatment with vaccination (*p* = 0.002).

## 4. Discussion

High milk feeding resulted in greater monocytes and eosinophils following the preweaning immune challenge, indicating a superior immune response in support of our first hypothesis (Figure 1 and Table 2). Monocytes are key immune cells that play a role in inflammation and provide protection against several forms of pathogens. This includes vaccination-induced immune responses, where increasing numbers of monocytes are recruited into the blood stream following vaccination of healthy individuals [36,37]. Eosinophils are known to increase in number during parasitic infections and allergic disease, providing protection to the host [38]. While the treatment differences in monocytes and eosinophils were as anticipated, the direction of the immune biomarker responses was not always as expected (i.e., reduction in eosinophils, neutrophils and basophils post vaccination); therefore, the acceptance of our first hypothesis comes with caution. Previous studies have found significant differences in immune competence between increased and restricted milk-fed calves in the preweaning phase [9,10], while others have not [12,13]. Discrepancies between results is not a new concept in calf nutritional studies, as many factors of management are interrelated, and all influence the health and physiological development of calves [39].

Differences in metabolic characteristics between treatments were previously reported in the companion paper of Ockenden et al. [28]. These results were as hypothesised, with the superior metabolic characteristics evident in the high-milk-fed calves in the preweaning phase. These were indicated by lower BHB; higher NEFA; and higher glucose, insulin and IGF-1. In terms of metabolic response, at 6 weeks of age in the current experiment, all the biomarkers responded to vaccination, but, like some of the immune biomarkers, not in the way we might expect. BHB reduced with vaccination for all treatment groups, when we would expect it to increase due to increased metabolic pressure [18,20] from the immune challenge. Similar anomalies occurred with glucose, insulin and IGF-1. Firstly, we could expect glucose concentrations to reduce with greater energy demand required for the immune response post vaccination [9]. However, this was not the case, and the subsequent increase in insulin for the low treatment group and decrease in insulin for the high treatment group with the corresponding increased glucose concentrations are further enigmatic. Regardless of glucose concentrations, insulin concentrations could also be expected to increase with vaccination for its suspected role in immunomodulation [40,41], which was only evident in the low treatment group, suggesting rejection of our hypotheses. Additionally, IGF-1 increased post vaccination for all treatment groups, when the metabolic status of the animal is expected to reduce as energy moves away from growth and towards the immune response [42]. IGF-1 is also closely related to insulin; therefore, the reduction in insulin with increased glucose and IGF-1 is further contradictory [43,44]. Aside from the abnormalities in trends or responses of the biomarkers, we can still conclude that the high-milk-fed calves (HH and HL) had overall superior metabolic characteristics at 6 weeks of age (as reported in Ockenden, et al. [28]). However, the actual responses of the immune challenge were not affected by treatment as anticipated.

The limited difference between preweaning treatment responses in the current experiment cannot be definitively explained but could come down to several potential factors. The first of these is the potential of interference from maternal antibodies still present in circulation of the calves from maternal vaccination in utero and via colostrum [45]. The vaccine used in this experiment is not novel, as it is routinely administered to cows prior to calving as part of normal farm management. Alternatively, as they had been previously exposed via maternal vaccination, the initial and most significant active immune response was potentially generated at a time when nutritional input was the same across treatments [46,47]. This previous exposure in utero and the presence of maternal antibodies prohibited the inclusion of a vaccine-specific antibody titre analysis to be included in this experiment. Secondly, the total energy intake from the high group could also be disadvantageous to the immune response, as an excessive level of nutrition can be as detrimental to immune competence as malnutrition [48]. The low-milk-fed calves in this experiment were also eating greater amount of concentrate; thus, their overall crude protein (CP) intake was similar to the high-milk-fed group (reported in Ockenden et al. [28]). Therefore, the nutritional treatments were not as different in terms of CP, which could account for some of the inconsistencies found between the similar management practices [9,28]. While the significantly greater monocyte and eosinophil counts in the high preweaning groups is suggestive of superior immune competence in the preweaning phase, the lack of difference in all other immune biomarkers and unexpected metabolic responses warrants further investigation for confirmation.

At 8 months of age, many of the immune biomarkers were not affected by the immune challenge at all, possibly suggesting a lack of response to the vaccination. Calves had received two previous doses of this vaccine since birth (6-week immune challenge and a booster at approximately 12 weeks of age, as per manufacturer’s instructions); hence, their humoral immune response may have reacted quicker, and the primary immune response may not have been as evident [46,47]. The eosinophil count was still significantly higher for the high-milk-fed heifers (HH and HL) than for the low-milk-fed heifers (LH and LL). While this could suggest some carry over effect in immunity from the preweaning treatment until 8 months of age, there was still a significant difference in bodyweight between the high and low-milk-fed calves, and it has previously been reported that calves with greater body condition have had greater eosinophil counts [49]. Monocytes were the only other immune biomarker to respond to the immune challenge; however, the differences previously detected between treatments in the preweaning phase were lost. Obeidat et al. [13] reported similar trends, whereby advantages in immune competence were prevalent from the level of nutrition in the preweaning phase, but the same effect was not observed in the immediate postweaning phase. However, contrary to our results, the reported advantages were evident in calves fed on the lower plane of nutrition in the preweaning phase by greater neutrophil activity. At 8 months of age, metabolic responses were mostly unaffected by treatment, and the few that were (insulin and QUICKI) were influenced by the nutritional input at the time, with no signs of developmental imprinting from the preweaning phase.

The inconsistencies in results continued at 13 months of age. The significant three-way interaction effects for the immune biomarker’s neutrophils, WBC and eosinophils are complicated. In the case of neutrophils and WBC, treatments with low preweaning nutrition (LH and LL) did not respond to the vaccination; however, those with high preweaning nutrition (HH and HL) did. Yet, the direction of this response was dependent on the postweaning nutrition, in that HH increased and HL decreased in WBC and neutrophil count post vaccination. These results are somewhat supported by previous work completed by this group in Ockenden et al. [16], where increased nutrition in the preweaning phase resulted in preserved superior immune response indicated by both WBC and neutrophil count post vaccination at 12 months of age. However, aside from the initial lack of differences in the preweaning phase of the current experiment, the opposite direction of this response between the HL and HH and the irregularity of the responses from the low preweaning treatment groups remain contradictory. Further, the eosinophil count was highest for the HL treatment group, which does not reflect the previously mentioned relationship between bodyweight trends and number of eosinophils [49]. Aleri, et al. [32] also found that calves with a greater bodyweight at 5–6 months of age had a superior immune responsiveness, using the vaccine toxoid specific antibody concentration as a measure, which was preserved when retested at 12–13 months of age. Given the eosinophil count was the only biomarker consistently different between treatments throughout the experiment, in combination with the lack of clear differences in immune competence in the preweaning phase, it is difficult to determine if developmental programming is occurring, and further, if accelerated postweaning growth rate can compensate for poor preweaning nutrition.

At 13 months of age, NEFA, glucose, insulin, IGF-1 and QUICKI were affected by postweaning treatment; however, like the results in the preweaning phase, they did not always respond as expected. As with the immune biomarkers, given the randomness of some of the responses, we are unable to conclude that preweaning nutrition is a critical phase of development for heifer metabolism, and it seems nutritional input at the time drives the responses rejecting our second hypothesis.

Health status or disease incidence was not formally recorded as a comparison of immune function as part of this experiment. However, any adverse health event is recorded as part of normal farm practice, and these were minimal throughout the entire experiment. While subjective, the reasonings for the low incidence of disease in even the LL treatment group is that this management mostly reflects the current Australian industry standard, which may still be fit enough to produce healthy calves that avoid disease.

It is also possible that, while statistically significant, the biological significance of these biomarker responses may be less important, given the immune changes within treatments are approximately 1 × 10^9^/L or less at 13 months of age (i.e., NEFA, glucose and QUICKI). The normal diagnostic limits for calves in the literature is limiting, and the literature available is not consistent and lacks animal numbers [50,51,52]. However, all the data presented in this paper were within normal diagnostic limits of at least one of these publications, further questioning the biological significance of the responses. However, these data contribute to the age-related changes in the biomarkers of healthy Holstein Friesian heifers.

While the data are suggestive of a positive influence of increased preweaning nutrition on the immune competence of dairy heifers, more research with a more novel vaccine, or variation in immune challenge type, would help to clarify the inconsistencies within this data set.

## 5. Conclusions

This experiment investigated the effects various pre- and postweaning rearing strategies had on the resilience of dairy replacement heifers via three immune challenges from birth until 13 months of age. The results outlined in this paper indicated a positive influence of increased preweaning nutrition on the immune competence of a calf, demonstrated by higher monocyte and eosinophil count in the high-milk-fed calves. To a lesser extent, the results also suggested potential preservation effects of preweaning nutrition on immune status at 13 months of age regardless of postweaning nutrition, demonstrated by eosinophil, WBC and neutrophil counts. However, these long-term advantages would need further clarification. The metabolic characteristics are influenced by the nutritional input at the time, regardless of any previously applied treatments. As discussed, a number of these biomarkers did not respond to the challenge as expected. While statistically significant, the biological significance of some of these results is negligible, and we caution against overinterpreting the data. More research using different immune challenge techniques or a more novel vaccination is necessary to confirm these findings.

## Figures and Tables

**Figure 1 animals-15-01379-f001:**
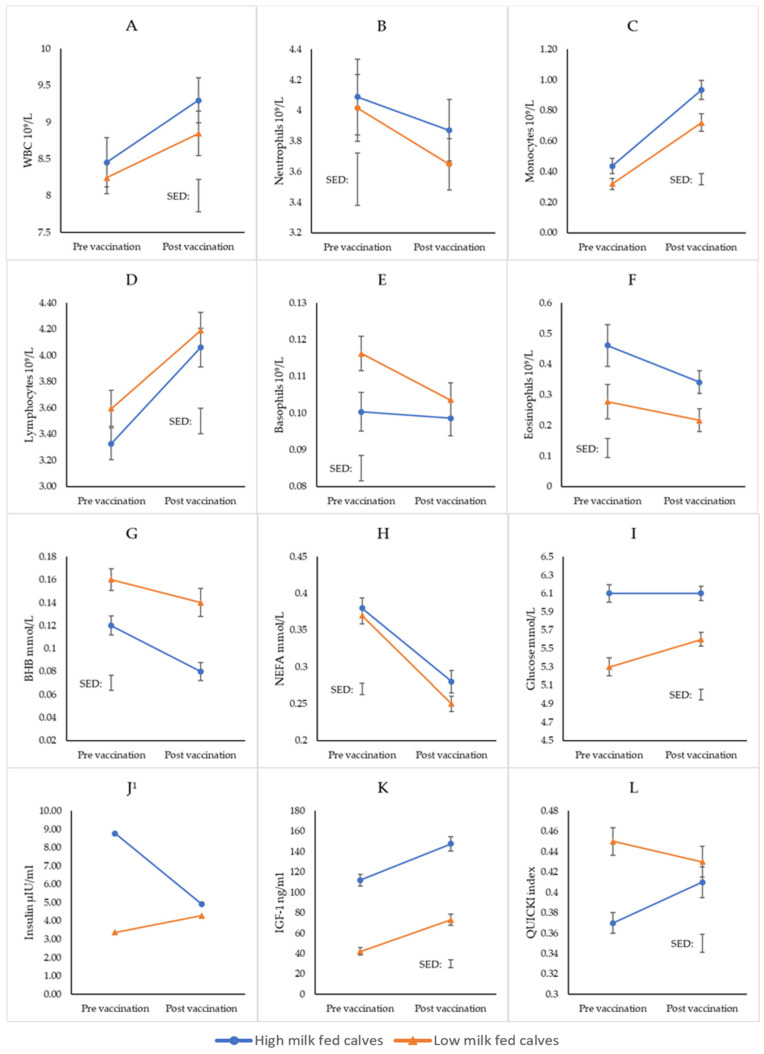
Immune and metabolic biomarkers (mean) for heifers fed either a high milk volume (8 L daily) or a low milk volume (4 L daily) at 6 weeks of age. (**A**): white blood cells (WBC); (**B**): neutrophils; (**C**): monocytes; (**D**): lymphocytes; (**E**): basophils; (**F**): eosinophils; (**G**): beta-hydroxybutyrate (BHB); (**H**): non-esterified fatty acid (NEFA) (**I**): glucose; (**J**): insulin; (**K**): insulin-like growth factor 1 (IGF-1); (**L**): quantitative insulin sensitivity check index (QUICKI)). The connecting line between pre- and post-vaccination indicates the direction of change, not the linear trend between points. Error bars indicate the standard error of the mean (SEM) on each data point, while floating error bars indicate the average standard error of the difference (SED). ^1^ Insulin was analysed in logarithmic scale, and the back-transformed means without their standard errors are presented.

**Figure 2 animals-15-01379-f002:**
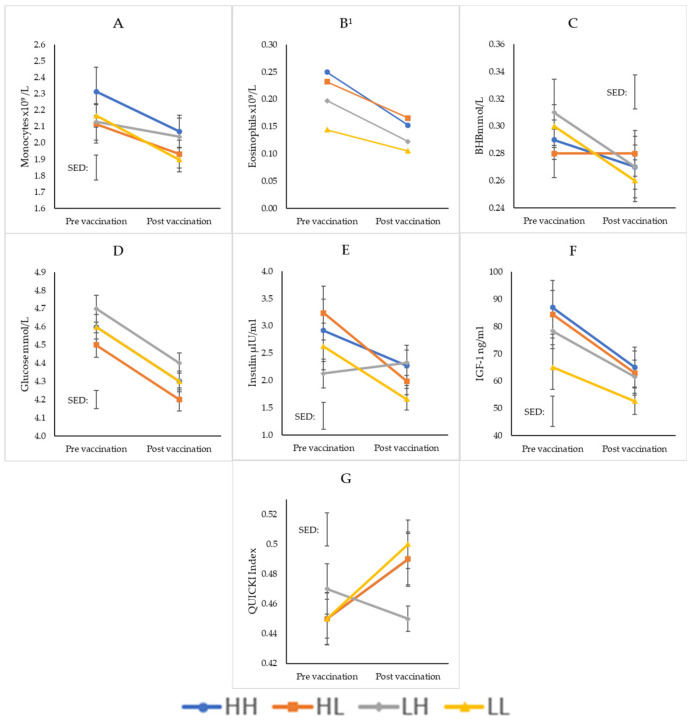
Immune and metabolic biomarkers (mean) for heifers on four different growth trajectories at 8 months of age (HH: high–high; HL: high–low; LH: low–high; LL: low–low). (**A**): monocytes; (**B**): eosinophils; (**C**): beta-hydroxybutyrate (BHB); (**D**): glucose; (**E**): insulin; (**F**): insulin-like growth factor 1 (IGF-1); (**G**): quantitative insulin sensitivity check index (QUICKI)). The connecting line between pre- and post-vaccination indicates the direction of change, not the linear trend between points. Error bars indicate the standard error of the mean (SEM) on each data point, while floating error bars indicate the average standard error of the difference (SED). ^1^ Eosinophils were analysed in logarithmic scale, and the back-transformed means without their standard errors are presented.

**Figure 3 animals-15-01379-f003:**
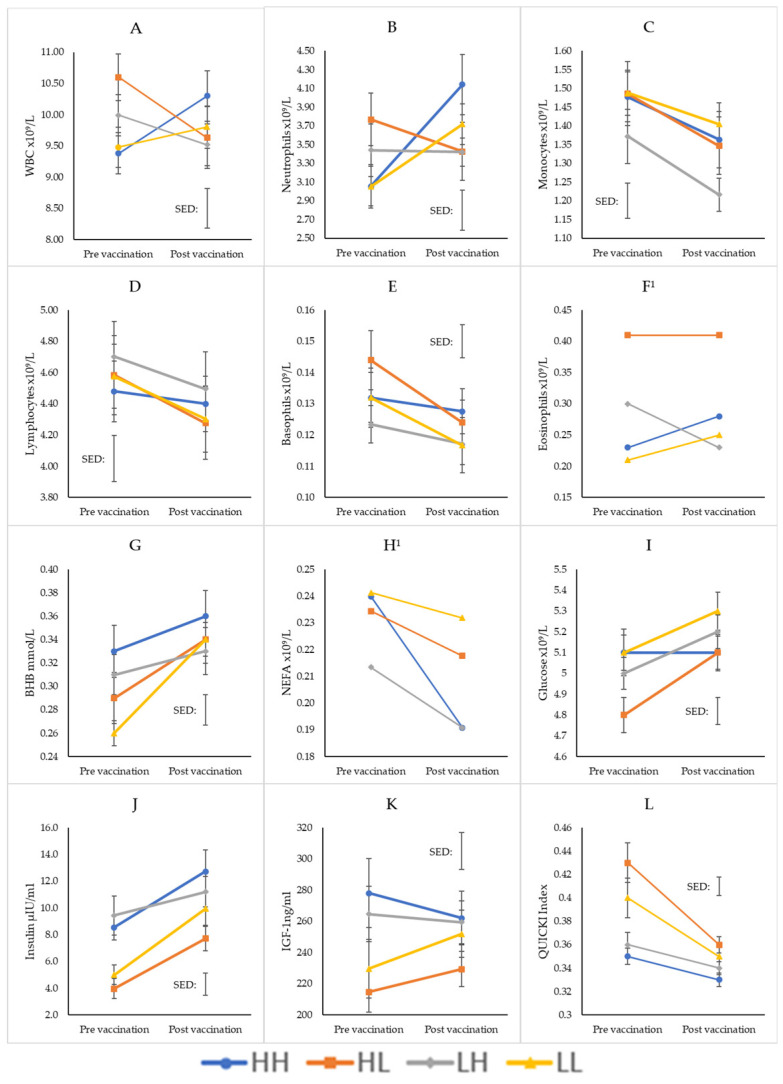
Immune and metabolic biomarkers (mean) for heifers on four different growth trajectories at 13 months of age (HH: high–high; HL: high–low; LH: low–high; LL: low–low). (**A**): white blood cells (WBC); (**B**): neutrophils; (**C**): monocytes; (**D**): lymphocytes; (**E**): basophils; (**F**): eosinophils; (**G**): beta-hydroxybutyrate (BHB); (**H**): non-esterified fatty acid (NEFA) (**I**): glucose; (**J**): insulin; (**K**): insulin-like growth factor 1 (IGF-1); (**L**): quantitative insulin sensitivity check index (QUICKI)). The connecting line between pre- and post-vaccination indicates the direction of change, not the linear trend between points. Error bars indicate the standard error of the mean (SEM) on each data point, while floating error bars indicate the average standard error of the difference (SED). ^1^ Eosinophil and NEFA values were analysed in logarithmic scale, and the back-transformed means without their standard errors are presented.

**Table 1 animals-15-01379-t001:** Mean bodyweights (kg) and concentrate intake (kg DM/heifer per week in the preweaning phase and kg DM/heifer per month in the postweaning phase) of heifers on four different growth trajectories at key timepoints in a pasture-based dairy system. Treatments: HH = high preweaning treatment and high postweaning growth rate; HL = high preweaning treatment and low postweaning growth rate; LH = low preweaning treatment and high postweaning growth rate; LL = low preweaning treatment and low postweaning growth rate. High preweaning treatment calves were fed 8 L of milk per calf per day, and low preweaning treatment calves were fed 4 L of milk per calf per day.

Time	Bodyweight (kg)				Concentrate Intake			
	HH	HL	LH	LL	HH	HL	LH	LL
Preweaning phase					(kg DM/heifer per week)			
Birth	38.7	39.8	38.7	39.1	0.00	0.00	0.07	0.07
6-week immune challenge	72.2	73.5	61.3	60.5	1.33	1.19	4.90	4.62
12 weeks (weaning)	117.8	119.0	106.2	103.1	9.80	9.52	12.18	12.18
Postweaning Phase					(kg DM/heifer per month)			
8-month immune challenge	244.9	239.6	232.8	222.7	1.1	0.49	10.9	0.0
13-month immune challenge	292.3	269.4	294.6	264.2	23.6	0.0	25.1	0.0
20 months (completion)	493.8	486.0	495.9	450.8	46.4	0.4	66.3	11.9

**Table 2 animals-15-01379-t002:** *p*-values for the main effects and interactions of immune and metabolic biomarkers for heifers on four different growth trajectories in response to immune challenges at 6 weeks, 8 months and 13 months of age (T1 = preweaning treatment; T2 = postweaning growth rate).

6-Weeks	*p*-Values								
	T1 Main Effect			Time Main Effect			T1 × Time		
WBC	0.421			<0.001			0.504		
Neutrophils	0.647			0.043			0.594		
Monocytes	0.017			<0.001			0.138		
Lymphocytes	0.273			<0.001			0.371		
Basophils	0.102			0.032			0.099		
Eosinophils	0.008			0.002			0.321		
BHB	<0.001			<0.001			0.201		
NEFA	0.042			<0.001			0.502		
Glucose	<0.001			0.022			0.079		
Insulin	<0.001			0.261			0.009		
IGF-1	<0.001			<0.001			0.603		
QUICKI	<0.001			0.339			0.008		
8-months	T1 main effect	T2 main effect	Time main effect		T1 × T2	T1 × Time		T2 × Time	T1 × T2 × Time
Monocytes	0.609	0.250	<0.001		0.543	0.744		0.567	0.256
Eosinophils	<0.001	0.252	<0.001		0.229	0.891		0.295	0.979
BHB	0.649	0.736	0.026		0.892	0.254		0.880	0.714
Glucose	0.091	0.258	<0.001		0.932	0.870		0.793	0.645
Insulin	0.169	0.904	<0.001		0.863	0.129		0.017	0.447
IGF-1	0.129	0.326	<0.001		0.519	0.380		0.766	0.807
QUICKI	0.879	0.597	0.002		0.526	0.367		0.023	0.056
13-months	T1 main effect	T2 main effect	Time main effect		T1 × T2	T1 × Time		T2 × Time	T1 × T2 × Time
WBC	0.483	0.828	0.805		0.625	0.894		0.184	0.002
Neutrophils	0.446	0.928	0.055		0.929	0.886		0.300	0.004
Monocytes	0.468	0.202	<0.001		0.180	0.923		0.735	0.495
Lymphocytes	0.702	0.674	<0.001		0.709	0.709		0.229	0.514
Basophils	0.171	0.533	0.002		0.986	0.844		0.081	0.633
Eosinophils	0.028	0.162	0.456		0.013	0.273		0.274	0.010
BHB	0.172	0.071	<0.001		0.588	0.679		0.258	0.364
NEFA	0.967	0.013	0.020		0.216	0.426		0.239	0.667
Glucose	0.106	0.960	<0.001		0.125	0.374		0.006	0.108
Insulin	0.535	<0.001	<0.001		0.308	0.657		0.315	0.182
IGF-1	0.760	0.030	0.508		0.396	0.436		0.016	0.894
QUICKI	0.660	<0.001	<0.001		0.199	0.557		0.002	0.472

## Data Availability

The raw data supporting the conclusions of this article will be made available by the authors on request. The data are not publicly available due to the experiment being conducted within Agriculture Victoria and are therefore bound by their policies.

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
