# Peer review of "Early Life Nutrition and Its Effects on the Developing Heifer: Immune and Metabolic Responses to Immune Challenges"

_animals, 2025, doi:10.3390/ani15101379_

Round 1
Reviewer 1 Report
Comments and Suggestions for Authors
The manuscript “Early life nutrition and its effects on the developing heifer: immune and metabolic responses to immune challenges”, ID animals-3540278, submitted to Animals as a research article, deals with immune competence, and metabolic characteristics studied via repeated vaccine immune challenges throughout early life of dairy replacement heifers.
According to the Journal aims, the paper deserves attention because of its potential relevancy in the resilience of dairy heifers. The topic is interesting, and this experiment is original, however the current paper (part B):
- does not sufficiently describe (in M&M) the essential results from the companion paper [28] on growth rates and feed intake of pre- and post-weaning nutritional strategies, involved in the discussion;
- forage to concentrate ratio of the diet should be reported and, because pasture was the majority of the diet, stocking rate and carrying capacity of pasture, its biomass production (kg DM/ha), and (total) heifers’ DM intake should be included in M&M to support discussion;
- models of statistical analyses are not reported by equations, for clarity;
- no results of statistics are reported in tables.
so that discussion on the metabolic responses of immune challenged heifers is not clear when related to the adopted nutritional strategies.
Author Response
AU: We would like to thank this reviewer for their time and constructive feedback on the manuscript. Their comments and suggestions have provided improvement in areas where the manuscript was lacking. We appreciate their contribution.
The manuscript “Early life nutrition and its effects on the developing heifer: immune and metabolic responses to immune challenges”, ID animals-3540278, submitted to Animals as a research article, deals with immune competence, and metabolic characteristics studied via repeated vaccine immune challenges throughout early life of dairy replacement heifers.
According to the Journal aims, the paper deserves attention because of its potential relevancy in the resilience of dairy heifers. The topic is interesting, and this experiment is original, however the current paper (part B):
- does not sufficiently describe (in M&M) the essential results from the companion paper [28] on growth rates and feed intake of pre- and post-weaning nutritional strategies, involved in the discussion;
AU: We thank the reviewer for this suggestion. Table 1 has been added to the Material & Methods section (L144-151) with key bodyweights and concentrate intakes summarised from the companion paper relevant to the current paper.
- forage to concentrate ratio of the diet should be reported and, because pasture was the majority of the diet, stocking rate and carrying capacity of pasture, its biomass production (kg DM/ha), and (total) heifers’ DM intake should be included in M&M to support discussion;
AU: We thank the reviewer for bringing this missing information to our attention. Unfortunately, individual pasture intakes were not able to be recorded for this experiment. However, a statement has been added to the materials and methods section to specify that pasture allocation was provided ad-libitum. L179-181: “Pasture was provided at what is considered ad-libitum (approximately 20kg/DM/cow per day). Individual pasture intakes were not recorded as part of this experiment.”
- models of statistical analyses are not reported by equations, for clarity;
AU: Thank you for your suggestion. The statistical model used in the analyses is now presented in equation form, along with a detailed description of each component (L252-266).
- no results of statistics are reported in tables.
AU: Again we would like to thank you for the suggestions and the appropriate P-values are included in Table 2.
so that discussion on the metabolic responses of immune challenged heifers is not clear when related to the adopted nutritional strategies.
Reviewer 2 Report
Comments and Suggestions for Authors
Material and methods
L118. How long has it taken to get all the 80 heifers? 4 months?
L 122. BPI is a genetic trail? Maybe put a short explanation for non-Australian readers.
- 145. Did all animals have brix above 9.4%? What do you mean by successful passive immune transfer?
L146. Milk from 2-4th milking is not colostrum. It is transitional milk. Colostrum is just the 1st milk!
L180. If you did a vaccination at 6 weeks of age, didn’t maternal immunoglobulins interfere? Clostridium vaccine recommendation would be after 12 weeks?
L182. It would be interesting to cite what is this vaccine for. Mainly for clostridiums.
Did you check for disease occurrence and health scores? It would be interesting to know.
Results
Figures 1, 2 and 3. Remove the horizontal bars from the graphs. Add error bars on the response and add an asterisk in the figure if there are statistical differences.
No tables?
I’ve missed the comparison of the biomarkers within time.
Author Response
AU: We would like to thank this reviewer for their time in reviewing our manuscript. Their comments have highlighted areas in the manuscript that were lacking, improving both the clarity and strength of discussion. We have responded to their points below.
L118. How long has it taken to get all the 80 heifers? 4 months?
AU: All 80 calves were born within 22 days. A statement has been added to L134-135 to provide this information: “All treatment allocations were full within 22 days.”
L 122. BPI is a genetic trail? Maybe put a short explanation for non-Australian readers.
AU: Thank you for bringing this oversight to our attention. More information has been added to L123-124 for clarity “including economic genetic traits such as production, health, fertility, longevity, workability and feed efficiency.”
L 145. Did all animals have brix above 9.4%? What do you mean by successful passive immune transfer?
AU: Thank you for identifying this important missing information. A statement has been added in the materials and methods to clarify the standard of successful passive transfer used in this experiment. L156-157: “Only calves with a serum Brix index concentration at or above 8.1% (equivalent to ≥ 10 g/L serum IgG concentration) were included in the experiment [31]”.
L146. Milk from 2-4th milking is not colostrum. It is transitional milk. Colostrum is just the 1st milk!
AU: This description has been rectified in the manuscript (L158). Thank you.
L180. If you did a vaccination at 6 weeks of age, didn’t maternal immunoglobulins interfere? Clostridium vaccine recommendation would be after 12 weeks?
AU: We agree and thank the reviewer for identifying this important point. The potential for interference from maternal immunoglobins is the reason analyses of the vaccine specific immunoglobins were excluded from this experiment and the calf’s innate immune response was used as a measure of immune competence. This justification has been expanded in the discussion (L416-419)
The first dose of the vaccination Ultravac 7in1 vaccination is recommended at 6 weeks of age with a second dose given at 12 weeks as per the manufacturer, Zoetis, protocol. Please find the link for further information your convenience: https://www2.zoetis.com.au/content/_assets/PDFs/L/Livestock-solution/zoetis-ultravac-7in1-product-information.pdf
L182. It would be interesting to cite what is this vaccine for. Mainly for clostridiums.
AU: A statement has been added L198-199 to provide more detail : “for protection against common leptospirosis and clostridial diseases”, we thank the reviewer for this suggestion.
Did you check for disease occurrence and health scores? It would be interesting to know.
AU: We would like to thank the reviewer for making this essential point. Disease occurrence was not something that was formally recorded as part of this experiment. However, as part as normal farm practice any adverse health incidence is recorded and was minimal throughout the experiment. A statement has been added to the discussion section to outline this. L480-485: “Health status or disease incidence was not formally recorded as a comparison of immune function as part of this experiment. However, any adverse health event is recorded as part of normal farm practice and was minimal throughout the entire experiment. While subjective, the reasonings for the low incidence of disease in even the LL treatment group is that this management most reflects the current Australian industry standard, which may still be fit enough to produce healthy calves that avoid disease”.
Results
Figures 1, 2 and 3. Remove the horizontal bars from the graphs. Add error bars on the response and add an asterisk in the figure if there are statistical differences.
AU: Thank you for these suggestions. The horizontal bars have been removed and have improved the readability of the figures. The SEM has also been provided in the error bars in the figures to provide more information. P-values have also been provided in the new Table 2 to easily identify the significant differences for main effects or the interactions.
No tables?
AU: P-value tables are now provided in Table 2.
I’ve missed the comparison of the biomarkers within time.
AU: The provided standard error of difference (SED) can be used to compare biomarkers within each time point and between time points. If the distance between any two treatment means at a given time or between times exceeds approximately 2×SED, the difference is considered statistically significant at the 5% level.
Reviewer 3 Report
Comments and Suggestions for Authors
File attached

Author Response
AU: We would like to sincerely thank the reviewer for their time in reviewing the manuscript. Their comments and suggestions have contributed greatly to the content provided and strengthened the manuscript.
This is very detailed write-up of part of a large experiment investigating the effect on the immune response of dairy heifers following two different pre-weaning and post-weaning nutritional regimes. It reads like a PhD programme and as a result is a bot over the top and unnecessarily long. This is partly due to some results that are difficult to justify and inconsistent.
There are a few details which would make this paper better. For example:
- Which of the two target pre-calving liveweight (600 or 530kg) is the 85% target mature weight recommended in Australia.
AU: The pre-calving 85% target mature liveweight in Australia is not explicitly specified, as it will vary considerably between herds/farms. These target weights are specific to Ellinbank Smart Farm and were determined using historical herd pre-calving body weight data.
- The difference between the two target pre-calving liveweights was only 12-13% (and less during the rearing phase). Is this a large enough difference to illicit treatment effects. A greater gap between the two target weight might have demonstrated greater effects.
AU: We thank the reviewer for this insightful comment. We agree that at times (particularly 8-month immune challenge when treatments were diverging) the current body weight differences may not have varied enough at the time to illicit any treatment effects. However, regardless of this, any preservation of the preweaning treatment effect may still have been able to be detected, or if any differences existed were a result of the current nutritional input.
- Although the difference in liveweight of the two pre-weaning treatments may have been given in a companion paper, they need to be presented here, particularly due to the authors comments about different concentrate intake of the two groups of calves. How big was the net effect on calf weaning weight?
AU: Thank you to the reviewer for this suggestion. Table 1 has been added to the Material & Methods section with key bodyweights and concentrate intakes summarised from the companion paper (L145-151). This suggestion has provided far more valuable information to be included to the manuscript in areas that were unclear.
- My main concern about this paper is the presentation of the results in Figures 1-3. It is incorrect to show these results as a straight line joining two points. It takes a minimum of three data points to justify the use of a straight line. These results should be presented in a table or histograms. These must be re-presented.
AU: In Figures 1–3, our intention was not to suggest a linear trend, but rather to illustrate the change in biomarker levels before and after the immune challenge. The line connecting the two-sampling occasion (pre and post vaccination) serves as a simple visual guide to help readers observe whether biomarker levels increased or decreased.
We fully agree that interpreting or fitting a linear trend requires at least three time points. However, that is not the purpose of our figures. We are simply indicating the direction of change between two defined sampling occasions (time points), which we believe adds clarity for the reader.
To avoid any confusion, we have revised the figure captions to clarify that the connecting lines do not represent a linear trend : “The connecting line between pre and post vaccination indicates the direction of change, not the linear trend between points”
I appreciate the authors' acknowledgement that some of the statistical differences they detected may not have biological significance. They could have specifically identified these.
AU: Thank you for this suggestion. The biomarkers have been listed in this sentence. L488-489: “It is also possible, that while statistically significant, the biological significance of these biomarker responses may be less important, given the immune changes within treatments are approximately 1 x 10⁹/L or less at 13- months of age (ie. NEFA; glucose and QUICKY).”
With correction of the presentation of the results this paper should be published.
Reviewer 4 Report
Comments and Suggestions for Authors
The paper »Early life nutrition and its effects on the developing heifer: Immune and metabolic responses to immune challenges« evaluated the effect of different pre- and postweaning nutrition on some parameters related to immune response to repeated vaccinations in heifers.
Four groups of heifers with different nutritional management were compared. Study is interesting, but to better evaluate the immune response, it would be useful to measure more parameters related to the immune response, such as different classes of antibodies, acute phase proteins, etc.
The title is suitable and clear enough.
The abstract is properly structured, reflecting the content of the article.
Introduction
The introduction is quite general. I suggest that the authors add information to the introduction about the importance and impact of nutrition on the immune system of calves and about biomarkers that can be used to assess the functioning of the immune system and the immune response to vaccination (e.g., immunoglobulins, etc.)
Materials and Methods
Lines 144-145 The authors state that they checked the appropriate passive immunity of the calves, I suggest that they add information about the limit value they used.
Lines 156-158 The authors state that the heifers had access to 3kg of concentrates per day. Was this amount available for an individual animal or for the entire group of animals in the box?
The authors say nothing about the health status of the animals during the experiment. I suggest that the authors add information about how they monitored the health status of the animals and if any of the animals diseased (diarrhea, pneumonia) during the experiment and how they were treated in this case. The health status of the animals has a significant impact on the functioning of the immune system and their immune response.
Lines 166-175 In the description of the post-weaning diet, it is not entirely clear how the diet differed between the individual experimental groups. I suggest that the authors clarify this further.
Line 209 The authors should add a whole word for Insulin-like growth factor 1 (IGF-1).
Results
When reporting the results, the authors could have added information on whether the heifers had any clinically detectable reactions to vaccination. Vaccination causes the formation of antibodies, so it would have been useful for the research if the authors had also measured the antibody titre in the blood of the calves after vaccination.
The figures in the results are informative, but it is not clear how large were variations in the values of the investigated parameters between individual animals (how dispersed the data were within the group). I suggest that the authors add this information to the figures. I think that boxplots would be more appropriate.
Discussion
In the discussion, the authors could comment on whether there were any differences in the health status of the animals between the groups. We would expect that animals with a better immune response would be healthier.
I cannot agree with the authors' claim in the discussion that a higher number of monocytes and eosinophils after vaccination means a better immune response, I think this is too general assessment.
I think that by measuring more specific parameters that indicate the functioning of the immune system, the authors would have come to better and more realistic findings, which could have been one of the conclusions of their study.
Author Response
AU: We would like to sincerely thank the reviewer for their time and constructive comments on the manuscript. Their suggestions have provided valuable information to be added to the manuscript to improve its clarity and strengthen the discussion. We appreciate their helpful feedback.
The paper »Early life nutrition and its effects on the developing heifer: Immune and metabolic responses to immune challenges« evaluated the effect of different pre- and postweaning nutrition on some parameters related to immune response to repeated vaccinations in heifers.
Four groups of heifers with different nutritional management were compared. Study is interesting, but to better evaluate the immune response, it would be useful to measure more parameters related to the immune response, such as different classes of antibodies, acute phase proteins, etc.
The title is suitable and clear enough.
The abstract is properly structured, reflecting the content of the article.
Introduction
The introduction is quite general. I suggest that the authors add information to the introduction about the importance and impact of nutrition on the immune system of calves and about biomarkers that can be used to assess the functioning of the immune system and the immune response to vaccination (e.g., immunoglobulins, etc.)
AU: We would like to thank the reviewer for their feedback. As discussed in a later response we agree the selection of immune biomarkers used in this experiment can be considered quite general, especially when considering the lack of vaccine specific antibody titre analyses (reasoning discussed below). The unfounded effect of nutrition on the biomarkers used in this experiment (white blood cells) is outlined in the introduction (L61-65), and we believe adding information regarding immune biomarkers that were not investigated may be misleading to readers. However, we are open to expanding on this topic and defer to the editor to decide if this additional information is required.
Materials and Methods
Lines 144-145 The authors state that they checked the appropriate passive immunity of the calves, I suggest that they add information about the limit value they used.
AU: We thank the reviewer for this suggestion. More information regarding the determination of passive transfer of immunity has been added to the manuscript. L156-157: “Only calves with a serum Brix index concentration at or above 8.1% (equivalent to ≥ 10 g/L serum IgG) were included in the experiment [31]”
Lines 156-158 The authors state that the heifers had access to 3kg of concentrates per day. Was this amount available for an individual animal or for the entire group of animals in the box?
AU: Thank you for identifying this error. This has been rectified in the manuscript L168: “3 kg/calf per day”.
The authors say nothing about the health status of the animals during the experiment. I suggest that the authors add information about how they monitored the health status of the animals and if any of the animals diseased (diarrhea, pneumonia) during the experiment and how they were treated in this case. The health status of the animals has a significant impact on the functioning of the immune system and their immune response.
AU: We thank the reviewer for identifying the lack of this valuable information. Disease incidence was not formally reported in this experiment, however, was recorded as per normal farm practice. Incidence of disease was minimal throughout this experiment and a statement has been added in the discussion outlining this. L480-485 “Health status or disease incidence was not formally recorded as a comparison of immune function as part of this experiment. However, any adverse health event is recorded as part of normal farm practice and was minimal throughout the entire experiment. While subjective, the reasonings for the low incidence of disease in even the LL treatment group is that this management most reflects the current Australian industry standard, which may still be fit enough to produce healthy calves that avoid disease”.
Lines 166-175 In the description of the post-weaning diet, it is not entirely clear how the diet differed between the individual experimental groups. I suggest that the authors clarify this further.
AU: An additional statement has been added in attempt to clarify this, thank you for bringing this to our attention. L186-188: “i.e. increasing concentrate allocation for treatment groups in instances where additional bodyweight gains were required achieve target weights.” The addition of Table 1 in the Materials and Methods section may also contribute to clarifying this.
Line 209 The authors should add a whole word for Insulin-like growth factor 1 (IGF-1).
AU: Thank-you for identifying this oversight. This has been corrected in the manuscript L226: “Insulin-like growth factor (IGF-1)”.
Results
When reporting the results, the authors could have added information on whether the heifers had any clinically detectable reactions to vaccination. Vaccination causes the formation of antibodies, so it would have been useful for the research if the authors had also measured the antibody titre in the blood of the calves after vaccination.
AU: We thank the reviewer for bringing up this important point. We agree that the inclusion of an antibody titre would have been an ideal analysis to include as in indication of response to the vaccination. However, we purposely omitted its inclusion due to the interference of maternal antibodies still present at the 6-week and potentially 8-month immune challenge confounding these results. This is briefly mentioned in the discussion section, and as a limitation of this experiment (further research to include a novel vaccine to avoid the confounding maternal antibodies is needed); however, we have expanded this description to highlight this importance. L416-418 “(rationalizing the exclusion of a vaccination specific antibody assay in this experiment). “This previous exposure in utero, and the presence of maternal antibodies prohibited the inclusion of a vaccine specific antibody titre analyses to be included in this experiment.”
The figures in the results are informative, but it is not clear how large were variations in the values of the investigated parameters between individual animals (how dispersed the data were within the group). I suggest that the authors add this information to the figures. I think that boxplots would be more appropriate.
AU: We appreciate the reviewer’s suggestion. In the current figures, we have included the standard error of the mean (SEM), which reflects the variability of the treatment mean estimates derived from the statistical model. While it is certainly possible to provide boxplots of the observed biomarker values at each sampling occasion to illustrate raw variability, doing so for all time points and treatments would result in a large number of plots, which may reduce readability and detract from the clarity of the main findings. We believe that the current presentation, using mean plots with SEM, appropriately conveys the results of the experiment in a statistically meaningful way.
However, we would be open to including boxplots for selected biomarkers at time points, should the reviewer or editor feel this would add substantial value to the presentation of the results.
Discussion
In the discussion, the authors could comment on whether there were any differences in the health status of the animals between the groups. We would expect that animals with a better immune response would be healthier.
AU: Thank you for identifying this important point. We agree that this information would be very valuable to the manuscript. Incidence of disease was not formally reported as part of the experiment but is recorded as normal farm practice and was minimal throughout the experiment and therefore was not included in our results. We have added a statement in the discussion section outlining this L480-485 “Health status or disease incidence was not formally recorded as a comparison of immune function as part of this experiment. However, any adverse health event is recorded as part of normal farm practice and was minimal throughout the entire experiment. While subjective, the reasonings for the low incidence of disease in even the LL treatment group is that this management most reflects the current Australian industry standard, which may still be fit enough to produce healthy calves that avoid disease”.
I cannot agree with the authors' claim in the discussion that a higher number of monocytes and eosinophils after vaccination means a better immune response, I think this is too general assessment. I think that by measuring more specific parameters that indicate the functioning of the immune system, the authors would have come to better and more realistic findings, which could have been one of the conclusions of their study.
AU: We thank the reviewer for their valued feedback. We agree that the lack of antibody titres in particular limits the conclusions regarding immune function in response to the vaccination immune challenge. While general, we still believe the preservation of these immune differences in eosinophils and monocytes as a result of preweaning nutrition provide a positive contribution to the lacking literature of the effects of nutrition on the longer-term immune status of dairy heifers. We acknowledge these concerns and will consider the inclusion of more immune biomarkers when planning and designing any future experiments.
Round 2
Reviewer 1 Report
Comments and Suggestions for Authors
Congratulations to the Authors.
Just at L. 180: 20 kg/DM/cow per day.
Author Response
Congratulations to the Authors.
Just at L. 180: 20 kg/DM/cow per day.
AU: We thank the reviewer again for their time and attention to detail when reviewing the manuscript. We have corrected this error. Their continued contribution has improved the quality of the manuscript.
Reviewer 2 Report
Comments and Suggestions for Authors
Congratulations for your work.
Author Response
Congratulations for your work.
AU: We thank the reviewer again for their time and valued feedback. Their contribution has greatly improved the manuscript.
Reviewer 4 Report
Comments and Suggestions for Authors
The authors' additional explanations have improved the manuscript considerably.
However, the authors should check the data on concentrate intake in Table 1, as it is strange that older heifers (8 months) would eat a smaller amount than younger ones. I suggest that they show the daily amounts of dry matter eaten per animal, both for heifers before weaning and after weaning. This will make the data easier to compare.
Author Response
The authors' additional explanations have improved the manuscript considerably.
AU: We thank the reviewer again for their contribution in improving the manuscript.
However, the authors should check the data on concentrate intake in Table 1, as it is strange that older heifers (8 months) would eat a smaller amount than younger ones. I suggest that they show the daily amounts of dry matter eaten per animal, both for heifers before weaning and after weaning. This will make the data easier to compare.
AU: We thank the reviewer for their valued feedback. We agree that older heifers would consume a greater total DM intake than younger heifers and how the information provided in Table 1 could first seem confusing. However, in this case, the amount of concentrate offered and consumed by the heifers at 8 months was less than during the preweaning phase. This experiment was conducted within a typical pasture-based system, where during the postweaning phase pasture was provided as majority of the diet. The amount of concentrate offered was supplied per experimental protocol as an additional supplement to influence the treatment average daily gains and only small amounts were required during this time. Unfortunately, individual pasture intakes were not able to be recorded for this experiment and therefore we are not able to provide daily total DM intakes postweaning. In attempt to clarify this we have adjusted the title of Table 1. L145-147 “Table 1. Mean bodyweights (kg) and concentrate intake (kg DM/heifer per week in the preweaning phase, and kg DM/heifer per month in the postweaning phase) of heifers on four different growth trajectories at key timepoints in a pasture-based dairy system.”